# The Potential of Using Generative AI/NLP to Identify and Analyse Critical Incidents in a Critical Incident Reporting System (CIRS): A Feasibility Case–Control Study

**DOI:** 10.3390/healthcare12191964

**Published:** 2024-10-02

**Authors:** Carlos Ramon Hölzing, Sebastian Rumpf, Stephan Huber, Nathalie Papenfuß, Patrick Meybohm, Oliver Happel

**Affiliations:** 1Department of Anaesthesiology, Intensive Care, Emergency and Pain Medicine, University Hospital Würzburg, Oberdürrbacher Str. 6, 97080 Würzburg, Germany; 2Psychological Ergonomics, University of Würzburg, 97070 Würzburg, Germany

**Keywords:** patient safety, healthcare quality improvement, human factors, human error, safety culture

## Abstract

Background: To enhance patient safety in healthcare, it is crucial to address the underreporting of issues in Critical Incident Reporting Systems (CIRSs). This study aims to evaluate the effectiveness of generative Artificial Intelligence and Natural Language Processing (AI/NLP) in reviewing CIRS cases by comparing its performance with human reviewers and categorising these cases into relevant topics. Methods: A case–control feasibility study was conducted using CIRS cases from the German CIRS-Anaesthesiology subsystem. Each case was reviewed by a human expert and by an AI/NLP model (ChatGPT-3.5). Two CIRS experts blindly assessed these reviews, rating them on linguistic quality, recognisable expertise, logical derivability, and overall quality using six-point Likert scales. Results: On average, the CIRS experts correctly classified 80% of human CIRS reviews as created by a human and misclassified 45.8% of AI reviews as written by a human. Ratings on a scale of 1 (very good) to 6 (failed) revealed a comparable performance between human- and AI-generated reviews across the dimensions of linguistic expression (*p* = 0.39), recognisable expertise (*p* = 0.89), logical derivability (*p* = 0.84), and overall quality (*p* = 0.87). The AI model was able to categorise the cases into relevant topics independently. Conclusions: This feasibility study demonstrates the potential of generative AI/NLP in analysing and categorising cases from the CIRS. This could have implications for improving incident reporting in healthcare. Therefore, additional research is required to verify and expand upon these discoveries.

## 1. Introduction

Critical Incident Reporting Systems (CIRSs) are widely recognised as a crucial component of patient safety in healthcare. In the 1999 Institute of Medicine’s report, “To Err is Human”, emphasised the need for incident reporting in healthcare, similar to the approach in aviation. The importance of incident reporting has been continuously reissued since [1,2]. Today, the analysis and utilisation of data from incident reporting systems not only aid in learning but are also pivotal in the proactive design of safer healthcare systems. These CIRSs serve as both a repository for capturing adverse events and a learning mechanism that enables continuous safety improvement. Incident reporting has become one of the key strategies endorsed by the World Health Organization (WHO), particularly highlighted in its Global Patient Safety Action Plan 2021–2030, where incident reporting is positioned as a cornerstone for advancing global patient safety initiatives [3]. CIRS success depends on several factors. One of the most important factors is providing feedback and acting on safety-critical aspects when collecting reports [4,5,6]. Failure to do so will negatively impact the quantity, diversity, and quality of reports. Various publications have cited the lack of direct feedback as a reason for underreporting [7,8].

The integration of Artificial Intelligence (AI) and Natural Language Processing (NLP) in healthcare has gained significant traction, particularly in enhancing diagnostic processes. Machine learning algorithms, particularly those leveraging deep learning techniques, have been employed to analyse medical imaging data, significantly enhancing the accuracy of disease detection [9]. Studies have shown that AI systems can achieve diagnostic accuracy comparable to that of experienced radiologists, particularly in fields such as oncology and pulmonary diagnostics [10,11]. Furthermore, the use of AI in diagnostic pathology has been highlighted as a transformative approach, with AI systems capable of identifying anomalies in histopathological images, thus streamlining the diagnostic workflow [12]. These advancements in AI highlight its potential to reshape various areas of healthcare, from diagnostics to administrative functions, including incident reporting systems. Recent studies further emphasise the pivotal role of AI in optimising clinical workflows and enhancing patient outcomes, particularly within diagnostic frameworks. AI-driven systems have demonstrated their capacity to not only improve diagnostic accuracy but also to integrate seamlessly into clinical processes, thereby increasing efficiency and supporting clinical decision making at a higher level of precision. Particularly, CIRSs stand to benefit substantially from AI and NLP integration [13].

While research on the application of AI and NLP in Critical Incident Reporting Systems (CIRSs) is still developing, several international studies have demonstrated the feasibility and potential of these technologies. A recent analysis of 6480 event reports from CIRSmedical.de demonstrated the utility of Natural Language Processing (NLP) in systematically analysing event reports and identifying patterns in reporting behaviour, sentiment, and keyword usage [14]. An analysis of the Swiss CIRRNET database confirms the value of NLP in the CIRS, using automated methods to identify key themes like medication errors and surgical complications [15]. The ability to categorise and synthesise incident reports using NLP can facilitate the identification of safety hazards and trends, ultimately leading to improved patient safety outcomes. Additionally, a systematic review has highlighted the utility of NLP for classifying incident reports and adverse events, further validating the applicability of these methods in improving patient safety [16]. By providing instant feedback on reported incidents, AI can enhance the engagement of healthcare professionals in the reporting process, addressing the common issue of underreporting due to lack of feedback. The aggregation of data from the CIRS through AI-driven analytics can also support healthcare organisations in developing targeted interventions to mitigate risks and enhance patient safety.

The main goals of this feasibility study are (a) to improve the review process of CIRS cases, (b) provide instant feedback on safety hazards and possibilities for improvement to the person reporting, (c) to categorise and classify critical incidents according to their relevance to the safety of patients, and (d) offer an aggregated synopsis of reported topics to the team of patient safety specialists responsible for the CIRS using generative Artificial Intelligence and Natural Language Processing (AI/NLP). This study aims to determine the feasibility and explore the impact of integrating AI/NLP into the CIRS. While the immediate goal is to assess the capabilities of AI in enhancing the efficiency and accuracy of CIRS processes, the broader ambition is to identify the key areas where AI can contribute to the development of a more robust and responsive patient safety framework. This study serves as a preliminary investigation, laying the groundwork for further research and development within our working group, as we continue to explore the potential of AI in advancing patient safety initiatives across healthcare systems.

## 2. Materials and Methods

This case–control feasibility study was conducted in December 2023. Two anaesthesiologists with special training in reviewing critical incident reports (CIRS experts) were presented with 12 CIRS cases, each with 1 review by an anaesthesiologist trained in analysing critical incident reports and 1 review by an AI/NLP model (ChatGPT 3.5, OpenAI Inc., San Francisco, CA, USA; 24 reviews in total). Both reviewers were experienced members of the CIRS evaluation committee (CIRS-AINS) and had many years of experience in evaluating CIRS cases. Furthermore, both reviewers possess a Master’s degree in Human Factors and System Safety from the University of Lund. Also, the two CIRS experts engaged with this subject matter from a scientific perspective.

The sample size of 24 reviews (12 human- and 12 AI/NLP-generated) was chosen for this feasibility study to provide a manageable and controlled comparison while maintaining the resources available for expert evaluation. The 12 cases were purposefully selected by the project team to exemplify typical CIRS cases, as well as typical entries that describe complaints rather than critical incidents. Specifically, two cases were complaints classified as CIRS cases, two cases involved technical difficulties, four were randomly chosen detailed CIRS reports, and four were randomly chosen brief CIRS reports. These cases were sourced from the CIRS-AINS portal (https://www.cirs-ains.de (accessed on 1 November 2023)), where cases are submitted anonymously. The CIRS reports used for all reviews are publicly available.

In order to find the prompt that generated the best results from ChatGPT, a number of prompts were trialled. After some careful prompt engineering, the exact prompt for generating individual reports was as follows: ‘In this chat, CIRS (Critical Incident Reporting System) reports from the anaesthesia department are displayed. Your task is to provide a brief and concise CIRS evaluation of the structural problems that have occurred for each case. You must not invent or omit any information. Organise your assessment according to the relevance of the problems that have arisen in the individual cases. Do not invent any new problems that were not explicitly mentioned in the case.’ Once the cases had been subjected to analysis, we instructed ChatGPT to provide a concise factual overview of the structural issues that had been identified in the aforementioned reports (Table 1). The exact prompt was as follows: ‘Now create a joint evaluation of all 11 CIRS (Critical Incident Reporting System) reports from the anaesthesia department that were given to you for evaluation in this chat. Your task is now to create a single short and concise CIRS evaluation of the structural problems that have arisen for all cases. You must not invent or omit any information. Organise your assessment according to the relevance of the problems that have arisen in the individual cases. Do not invent new problems that were not explicitly mentioned in the case.’ The original reports and their generated analyses were not subjected to any form of pre- or post-processing.

A standardised online questionnaire was utilised to present the cases and corresponding reviews to the CIRS experts in a randomised order. The experts were blinded to the method of generation employed for each review. Their only information regarding the generation process was that half of the 24 reviews (12 cases) were generated by a human and half by the AI/NLP model. The evaluation of the semantic and syntactical quality was based on four criteria: (1) quality of linguistic expression, (2) recognisable expertise, (3) logical deducibility, and (4) overall quality. There was no visible tracking available to the experts during the trial to indicate how many reviews they had already assigned to each condition (human vs. AI/NLP). Furthermore, the reviews were presented sequentially, and the experts were not allowed to return to previously assessed reviews. The questions were to be answered on a 6-point Likert scale, which was based on the German school grading system that was familiar to both CIRS experts. The scale ranged from 1 to 6 (1 = very good, 2 = good, 3 = satisfactory, 4 = adequate, 5 = inadequate, 6 = insufficient). Additionally, we requested the assumed method of generation of the reviews, which was to be answered dichotomously, indicating whether the method of generation was human or AI/NLP. The level of confidence that the CIRS experts had in their decision regarding the method of generation of the ratings (human vs. AI/NLP) was represented by a slider that could be adjusted between 0% and 100% in 10% increments with no default value.

Quantitative data are reported with mean ± standard deviation. The Mann–Whitney U test was employed for statistical comparison due to its suitability for non-normally distributed data and its ability to handle ordinal data from independent groups. The *p*-value for all statistical tests was set at *p* ≤ 0.05. We adhered to the Strengthening the Reporting of Observational Studies in Epidemiology (STROBE) checklist [17]. Given the observational nature of this study, the checklist was used to guide the reporting of essential elements. A completed version of the STROBE checklist is provided as a Appendix A. Additionally, example reviews translated into English are listed in the Appendix A.

## 3. Results

Of the 12 case reviews performed by a human, both CIRS experts attributed 11 to humans and 1 case analysis falsely to Artificial Intelligence. CIRS expert #1 classified 6 of the 12 AI/NLP reviews (50%) as human. These six AI/NLP-created reviews were suspected as human with a mean confidence of 76 ± 32.6% by CIRS expert #1. CIRS expert #2 classified 5 of the 12 AI/NLP-created reviews (41.6%) as human. The five misclassified reviews were rated as human with an average confidence of 98 ± 4.47%. In terms of topic categorisation, the AI/NLP model was able to independently categorise the cases into relevant themes. The categorisation and summary of cases can be seen in Table 1. Figure 1 visually presents the comparative statistical analysis of the ratings for human- and AI/NLP-generated reviews across the key dimensions: linguistic expression, recognisable expertise, logical derivability, and overall quality. In terms of linguistic expression, the human-created reviews were on average rated at 3.13 ± 0.97, descriptively inferior to the AI/NLP-created reviews at 2.92 ± 0.97 (*p* = 0.39, U = 249.0). The average score for recognisable expertise was 2.83 ± 0.81 in the human- and 2.83 ± 1.12 (*p* = 0.89, U = 282.0) in the AI/NLP-created reviews. The dimension of logical derivability received an average score of 2.79 ± 0.83 in the human- and 2.96 ± 1.23 (*p* = 0.84, U = 297.0) in the AI/NLP-created reviews. The average overall quality of the human-created reviews was rated at 3.17 ± 1.09 and that of the AI/NLP-created reviews was rated at 3.12 ± 1.93 (*p* = 0.87, U = 280.0), respectively.

## 4. Discussion

The results show that both CIRS experts attributed most human-created case reviews correctly, with only 4.2% of case reviews falsely classified as AI/NLP-created. On average, the CIRS experts incorrectly classified 45.8% of AI/NLP-generated reviews as human-generated. These results underline the difficult to impossible distinction between the two methods of generation of the CIRS reviews and thus the high quality of the AI/NLP-created CIRS case reviews. Studies indicate that AI systems can perform clinical tasks with a precision that rivals or surpasses that of human practitioners. For example, the literature suggests that AI can execute diagnostic tasks with a speed and accuracy that may exceed human capabilities, particularly in pattern recognition and data analysis [18]. While this demonstrates the potential of AI to streamline the review process, it also introduces potential risks. In healthcare, the perception of AI as a “black box”—where the decision-making process is not transparent—can hinder its acceptance among healthcare professionals [19].

Furthermore, the generative AI/NLP model demonstrated the capability to categorise cases into relevant topics independently. This automatic categorisation could serve as a potential application for AI in the management of critical incident reports, facilitating the more efficient sorting and prioritisation of incidents based on their urgency and relevance to patient safety. The ability of AI to quickly categorise large datasets into meaningful themes could revolutionise the triaging process in incident reporting systems, allowing healthcare institutions to allocate resources more effectively. The implications of this are profound, as effective categorisation can facilitate quicker responses to incidents and enhance overall healthcare delivery efficiency. AI systems excelled in categorising cases into relevant themes, showcasing their ability to process and analyse large datasets efficiently. AI’s categorisation within healthcare can be observed in its diverse applications, which include diagnostic imaging, patient monitoring, and personalised treatment plans. For instance, Kłoska et al. discuss the use of AI models in classifying chest X-rays, demonstrating that AI can achieve performance levels comparable to human radiologists [19]. This capability aligns with findings from previous studies that have noted AI’s strength in handling structured data and identifying patterns that may not be immediately apparent to human reviewers [20].

Remarkably, the average linguistic quality of the AI-created reviews (2.92) was valuated slightly, although not significantly, better than that of the human-created reviews (3.13). Consistent with our results, O’Neill et al. showed that students’ spelling and grammar improved significantly when they used the AI program Grammarly [21]. The integration of specialised AI models trained on biomedical datasets, such as BioGPT, has been suggested to enhance the linguistic quality of AI-generated content in healthcare [22]. The recognisable expertise was comparable between the groups, with an average rating of 2.83. These results may be due to several factors. Firstly, only easy-to-analyse CIRS cases were selected. Secondly, the analysis of the CIRS cases was only carried out with key points. A real analysis is typically written out in full text. Future studies should consider incorporating more complex, nuanced cases to further investigate the limitations and capabilities of AI/NLP models in healthcare settings. The logical derivability of the human-created reviews was similar (2.79) to the AI/NLP-created ones (2.96). Concerning the overall quality of the reviews, human-created reviews were again comparable (3.17) to AI/NLP-created reviews (3.12). Additionally, improving the linguistic quality would enhance the overall impression. Moreover, the successful implementation of AI and NLP in healthcare hinges on addressing several challenges, including data privacy concerns, ethical considerations, and the need for robust regulatory frameworks to ensure the safe deployment of AI technologies [23].

While our results demonstrate the impressive capabilities of AI, they also indicate potential risks. For instance, AI systems might eventually be relied upon too heavily, leading to a reduction in critical oversight by human experts, who may become less vigilant when reviewing AI outputs. Moreover, the inability to easily distinguish AI from human reviews could raise concerns about accountability in healthcare settings, where the origin of safety-critical evaluations must be clear. If AI-generated reviews contain errors, it may be difficult to trace responsibility, which could impact patient safety and legal accountability. Furthermore, the implications of algorithmic bias and the ethical responsibility of AI developers must be considered, as biases inherent in AI systems could perpetuate existing disparities in healthcare delivery [24].

In terms of practical implementation, the integration of AI/NLP into CIRS management presents both opportunities and challenges. On the one hand, the efficiency and consistency with which AI models can process large volumes of data offer clear advantages for incident reporting systems that are often resource-constrained. On the other hand, the ethical implications of deploying AI in such sensitive areas demand careful consideration. Issues such as data privacy, bias in AI models, and the potential for diminished human oversight must be addressed within robust regulatory frameworks to ensure that AI technologies are used responsibly in healthcare. The deployment of AI/NLP technologies should not be viewed as a replacement for human expertise but rather as a complement, enhancing the ability of human reviewers to focus on the more complex and context-sensitive aspects of incident analysis.

A technical limitation within our approach was using an off-the-shelf general AI model. While we used carefully engineered prompts, fine-tuning the model with technical terms from healthcare would likely enhance its capabilities [25]. For future research, it is recommended to explore the integration of AI/NLP with other machine learning techniques, such as reinforcement learning or supervised learning models, to improve the AI’s deductive reasoning capabilities. Moreover, conducting longitudinal studies to assess the long-term impact of AI-driven reviews on patient safety would provide valuable insights into how these systems evolve and integrate into clinical workflows over time. In future studies, a higher number of experts, a greater number of CIRS cases, and other AI/NLPs or specially trained AI/NLPs should be considered to support our findings. Additionally, exploring the potential of hybrid AI–human review models, where AI assists in categorisation and initial analysis, and human experts provide final contextual evaluations, could offer the best of both worlds. Such human-in-the-loop systems are referred to as “Useful AI” and could ensure efficiency while maintaining the essential human oversight needed in patient safety systems.

The promising results of our study highlight the potential of AI/NLP technologies in enhancing the quality and efficiency of critical incident report management. Building on our exploratory investigations, we aim to develop an intelligent CIRS system and evaluate it multicentrically.

## 5. Conclusions

This study demonstrates that generative AI, using a Natural Language Processing approach, can be used to generate reviews of CIRS cases hardly distinguishable from human-written reviews. Furthermore, our hypothesis that the quality of the CIRS case reviews—assessed in terms of linguistic quality, recognisable expertise, logical derivability, and overall review quality—of AI/NLP-created reviews would be non-inferior to human-created analyses was confirmed by the study results. Additionally, the AI’s ability to automatically categorise cases into relevant topics could serve as a useful tool in the management of critical incident reports. This could have important implications for further improving incident reporting in healthcare.

## Figures and Tables

**Figure 1 healthcare-12-01964-f001:**
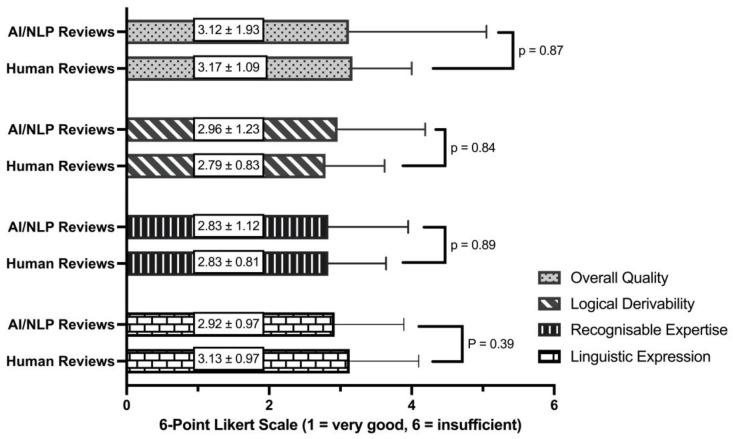
Comparison of human- and AI/NLP-generated reviews across key dimensions (mean ± SD).

**Table 1 healthcare-12-01964-t001:** Categorised CIRS reviews by ChatGPT.

Case Number	CIRS Review
102299, 118493, 128255, 247077, 252263	Lack of communication or information transfer between different departments of the hospital
102299, 118493, 128255, 247077, 252263	Lack of standardisation of procedures and processes, especially with regard to checklists and protocols
102299, 118493, 128255, 252263	Lack of availability or accessibility of important medical equipment, instruments or medication
102299, 118493, 252263	Inadequate review or verification of information, especially in the preparation of patients or the collection of relevant data
102299, 128255, 252263	Lack of compliance with established safety measures and guidelines, particularly in relation to medical procedures and processes
102299, 118493, 247077	Problems in dealing with staff shortages and time constraints, especially in emergency situations or at peak times
102299, 118493	Insufficient training or knowledge of staff regarding certain medical procedures or guidelines
102299, 118493	Missing or incomplete documentation of important information in patient medical records or clinical notes
102299, 252263	Lack of or inadequate preparation for rare or unusual medical situations, particularly related to blood transfusions or special patient needs
252263	Problems with the availability or compatibility of medical devices or instruments, in particular in connection with the use of certain brands or models
252263	Lack of arrangements to accommodate patient preferences or requirements, particularly in relation to specialised medical treatments or procedures

## Data Availability

The data presented in this study are available on request from the corresponding author.

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
