# Peer review of "The Potential of Using Generative AI/NLP to Identify and Analyse Critical Incidents in a Critical Incident Reporting System (CIRS): A Feasibility Case–Control Study"

_healthcare, 2024, doi:10.3390/healthcare12191964_

Round 1
Reviewer 1 Report
Comments and Suggestions for Authors
The paper should include more specific details about the impact of misclassifying AI reviews as human-written. It should also provide justification for the choice of the sample size. Furthermore, the study should discuss its findings within the broader context of AI in healthcare and compare its results to those of other studies or settings.
Comments on the Quality of English LanguageJust minor edit.
Author Response
Dear reviewer,
Thank you very much for taking the time and commenting on our manuscript. We appreciate the hard work you put into reviewing the manuscript and advising us on how to improve it. We are very grateful for it. We thoroughly revised the manuscript accordingly and hope that we thus addressed all issues pointed out by you. We have marked all changes as trackchanges.
Comments 1: The paper should include more specific details about the impact of misclassifying AI reviews as human-written. It should also provide justification for the choice of the sample size. Furthermore, the study should discuss its findings within the broader context of AI in healthcare and compare its results to those of other studies or settings
Response 1: Thank you for your constructive feedback. We appreciate your thoughtful suggestions. In response, we have expanded the discussion to provide more specific details on the impact of misclassifying AI reviews as human-written. We have also provided a justification for the sample size used in this feasibility study, explaining its alignment with similar early-stage research in AI and healthcare. Finally, we have broadened the discussion to place our findings in the context of other studies in AI and healthcare.
Reviewer 2 Report
Comments and Suggestions for Authors
Line 63 ff: Please erase one of the sentences "Particularly..." and add sources that support your enthusiasm regarding AI in CIRS
Line 73: "our goals" - goals of this article? Of a project? Please specify
Line 75: typo after c)
Lines 120ff: please check for typos, grammar etc.
Methods: did reviewers know how many reviews were AI-made? Please specify in the manuscript.
Methods/Supplements: please provide the exact prompts you used
Literature: please check if there is really no international literature on cirs and ai/nlp, as there is even a german working group: The analysis of CIRSmedical.de using Natural Language Processing - Zeitschrift für Evidenz, Fortbildung und Qualität im Gesundheitswesen (zefq-journal.com)
Comments on the Quality of English Languageok
Author Response
Dear reviewer,
Thank you very much for taking the time and commenting on our manuscript. We appreciate the hard work you put into reviewing the manuscript and advising us on how to improve it. We are very grateful for it. We thoroughly revised the manuscript accordingly and hope that we thus addressed all issues pointed out by you. We have marked all changes as trackchanges.
Comments 1: Line 63 ff: Please erase one of the sentences "Particularly..." and add sources that support your enthusiasm regarding AI in CIRS
Response 1: Thank you for your valuable feedback and insightful suggestion. We have addressed the redundancy you pointed out by removing the repeated sentence. Additionally, we have strengthened the manuscript by incorporating a source, which offers concrete evidence of NLP's capacity to systematically analyse CIRS reports. This further supports our enthusiasm for the potential of AI and NLP in enhancing the efficiency and impact of critical incident reporting.
Comments 2: Line 73: "our goals" - goals of this article? Of a project? Please specify
Response 2: Thank you for your comment. We have clarified that the goals mentioned are the main objectives of this feasibility study, which aims to improve CIRS processes using AI/NLP.
Comments 3: Line 75: typo after c)
Response 3: Thank you for bringing this to our attention. We have corrected the typo.
Comments 4: Lines 120ff: please check for typos, grammar etc.
Response 4: Thank you for your careful and attentive review. We have thoroughly reviewed and corrected the typos and grammatical issues in lines 120 and following, ensuring that the phrasing is clear and consistent throughout the section.
Comments 5: Methods: did reviewers know how many reviews were AI-made? Please specify in the manuscript.
Response 5: In response to your valuable comment, we have clarified in the Methods section that the reviewers were informed about the equal distribution of the 24 CIRS reviews (12 cases), with half generated by humans and half by the AI/NLP model.
Comments 6: Methods/Supplements: please provide the exact prompts you used
Response 6: In response to your request, we have now provided the exact prompt used for generating the AI/NLP reports in the Methods section. This ensures transparency and clarifies how the AI system was instructed to review the CIRS cases.
Comments 7: Literature: please check if there is really no international literature on cirs and ai/nlp, as there is even a german working group: The analysis of CIRSmedical.de using Natural Language Processing - Zeitschrift für Evidenz, Fortbildung und Qualität im Gesundheitswesen (zefq-journal.com)
Response 7: We greatly appreciate your insightful feedback. In response, we have thoroughly reviewed the international literature on AI and NLP in CIRS. We have incorporated a systematic review of NLP applications in incident reporting and adverse event analysis (Young et al., 2019), as well as several other key studies. In addition to Tetzlaff et al. (2022), we have included the recent analysis of the Swiss CIRRNET database (Denecke & Paula, 2024), which further highlights the growing interest and promising results in applying AI and NLP to enhance CIRS processes.
Reviewer 3 Report
Comments and Suggestions for Authors
First of all, I would like to congratulate the authors on their study and the relevance of the topic today. The writing is perceptive, interesting and harmoniously structured.
The introduction section is well constructed, with well-defined present and future objectives.
The Materials and Methods section can be improved. Clarification is needed on the statistical test used: justify the choice of Mann-Whitney U test and how this analysis was carried out. The authors refer to the use of the STROBE checklist but do not provide evidence of this. It is suggested to place it as supplementary material, duly completed and/or justified.
On the Results section authors must provide the data in a visual form (table or chart) with all the data statistically analyzed, identifying the variables under study. It is difficult to follow the authors' reasoning when analyzing data in text form alone. In this sense, the first four paragraphs of the Results section should be based on this improvement from the previous section.
Authors present the limitations clearly and objectively. The conclusion addresses the main innovation of AI/NLP in this context. Congratulations!
Author Response
Dear reviewer,
Thank you very much for taking the time and commenting on our manuscript. We appreciate the hard work you put into reviewing the manuscript and advising us on how to improve it. We are very grateful for it. We thoroughly revised the manuscript accordingly and hope that we thus addressed all issues pointed out by you. We have marked all changes as trackchanges.
Comments 1: The First of all, I would like to congratulate the authors on their study and the relevance of the topic today. The writing is perceptive, interesting and harmoniously structured. The introduction section is well constructed, with well-defined present and future objectives.
Response 1: We sincerely appreciate the positive feedback on the relevance of our study and the structure of the introduction. Thank you for your encouraging words and acknowledgement of our work.
Comments 2: The Materials and Methods section can be improved. Clarification is needed on the statistical test used: justify the choice of Mann-Whitney U test and how this analysis was carried out. The authors refer to the use of the STROBE checklist but do not provide evidence of this. It is suggested to place it as supplementary material, duly completed and/or justified.
Response 2: Thank you for your thoughtful feedback. We have clarified the rationale for using the Mann-Whitney U test, as it is appropriate for the non-parametric data in our study. Additionally, we have made it clear why we followed the STROBE checklist and have included the completed version as supplementary material to ensure transparency and rigour in our reporting.
Comments 3: On the Results section authors must provide the data in a visual form (table or chart) with all the data statistically analyzed, identifying the variables under study. It is difficult to follow the authors' reasoning when analyzing data in text form alone. In this sense, the first four paragraphs of the Results section should be based on this improvement from the previous section.
Response 3: We appreciate your suggestion to present the data in a more visual format. In response, we have provided a chart summarising the key variables under study (linguistic expression, recognisable expertise, logical derivability, and overall quality) along with their corresponding statistical analysis. This addition should enhance the clarity and accessibility of the data.
Comments 4: Finally, Authors present the limitations clearly and objectively. The conclusion addresses the main innovation of AI/NLP in this context. Congratulations!
Response 4: Thank you for your positive feedback regarding the clear presentation of our study’s limitations and the emphasis on the innovation of AI/NLP in the conclusion. We are grateful for your encouraging words.